# Exploring Patients' Perspectives on Late Complications after Colorectal and Anal Cancer Treatment: A Qualitative Study

**Birthe Thing Oggesen [1],*, Marie Louise Sjødin Hamberg [1], Thordis Thomsen [2,3] and Jacob Rosenberg [1]**

1 Department of Surgery, Herlev Hospital, University of Copenhagen, DK-2730 Herlev, Denmark; marie.louise.sjoedin.hamberg.01@regionh.dk (M.L.S.H.); jacob.rosenberg@regionh.dk (J.R.)
2 Department of Anaesthesiology, Herlev Hospital, University of Copenhagen, DK-2730 Herlev, Denmark; thordis.thomsen@regionh.dk
3 Department of Clinical Medicine, University of Copenhagen, DK-2200 København, Denmark
* Correspondence: birthe.thing.oggesen@regionh.dk; Tel.: +45-38682412

**Abstract:** Background: Patients often experience late complications following treatment for colorectal and anal cancer. Although several measurement tools exist to classify the severity of these symptoms, little is known about how patients personally experience and adapt to these complications. This study aimed to investigate patients' experiences and coping strategies in relation to these symptoms. Methods: We conducted an explorative qualitative interview study to gather data. Results: Our findings revealed two main categories: How patients react after treatment for colorectal and anal cancer, and Experienced symptoms. Additionally, we identified four sub-categories: the period after discharge, coping strategies, stool symptoms, and other symptoms. Patients commonly feel abandoned once their surgical and oncological treatments are completed. It is typical for patients to turn to the internet for guidance on managing late complications, despite being aware that evidence-based options are limited. Stool-related issues significantly impact patients' personal and professional lives, requiring constant preparedness for accidents, the use of diapers, and the need for extra clothing at all times. Furthermore, patients experience additional troublesome symptoms such as urinary incontinence, fatigue, pain, and sexual dysfunction, which further affect their daily lives. Conclusions: Patients experience multiple problems after colorectal cancer surgery, and this warrants more focused attention.

**Keywords:** colorectal cancer; late complications; anal cancer; survivorship

## 1. Introduction

Patients treated for colorectal and anal cancer are at high risk of developing late complications following oncological and surgical treatment [1]. The most common late complications are stool-related symptoms [2]. Patients treated for colon cancer, both right and left sided, have a 31% risk of experiencing incontinence for stool and a 33% risk of difficulties in rectal emptying [3]. Additionally, they have a 21% risk of experiencing major Low Anterior Resection Syndrome (LARS) [4,5]. After rectal cancer surgery, up to 69% of patients may experience major LARS [4,6,7]. Another common late complication is sexual dysfunction. The incidence of sexual dysfunction appears to be higher in men compared to women. For example, erectile dysfunction is reported in 54% of patients treated for rectal cancer, while the risk of vaginal dryness is reported in 35% of patients. Other sexual dysfunctions may include dyspareunia and a lack of arousal or sexual desire [2,8]. Urinary problems such as stress, urge, and overflow incontinence following rectal cancer surgery and chemo-radiation therapy for anal cancer are also frequently reported, with an incidence up to 70% among patients [2,9–11]. Other well-known but less investigated late complications include chemo-induced peripheral neuropathy caused by Oxaliplatin [2], persistent tiredness or fatigue, which can greatly impact the patients [2], and increased psychological distress due to the cancer diagnosis and the treatment [2,12]. Existing scoring

tools, such as LARS [5], can help estimate and assess the severity of these symptoms [1]. However, there is limited knowledge regarding how these late complications impact patients' daily lives and how they cope with them. Understanding these aspects is crucial for planning effective interventions in the future.

This qualitative study aims to explore how patients, in their own words, describe the experience of living with late complications and the coping mechanism they employ.

## 2. Materials and Methods

We conducted a qualitative study and reported the results in accordance with the Consolidated Criteria for Reporting Qualitative research (COREQ) guidelines [13].

### 2.1. Research Team and Reflexivity

The research team comprised two interviewers (MSH (RN) and TT (RN, Prof.)), two analysts (TT and BTO (MD)), and three individuals involved in conceptualization (MSH, BTO, and JR (MD, Prof.)). Both interviewers were registered nurses, and one of them (TT) also held a professorship. Both interviewers were female. One interviewer (TT) had extensive experience in qualitative research, including conducting interviews, while the other (MSH) was less experienced but received training from a highly skilled individual. The interviewers had no prior relationship with the participants, although one interviewer worked as a scientist in the Late Complication Clinic from which the participants were recruited. The participants were informed by the interviewer about the interviews and provided written information before agreeing to participate.

### 2.2. Study Design

The methodological orientation employed a hermeneutic approach, combining deductive and inductive methods through content analysis. The study framework adopted a descriptive approach that focused on capturing patients' perceptions and expressions of living with late complications after cancer treatment.

Participants were selected through purposeful sampling [14], taking into account factors such as age, different symptoms (based on records from the Late Complication Clinic), and gender. All participants had received treatment either for colorectal or anal cancer and had attended the Late Complication Clinic at Herlev Hospital. Recruitment was conducted either face-to-face or over the phone. All surgical patients were offered follow-up in the surgical outpatient clinic 10 days, 6 months, and 36 months after surgery for clinical control with a surgeon. Furthermore, they were contacted by a nurse every half year with alternating answer of CT scan or CEA measurement between 6 and 36 months.

Data were collected through telephone interviews due to the COVID-19 pandemic. The interviews were audio-recorded and later transcribed verbatim. No field notes were taken. The interviews took place at participants´ homes or their preferred location, with only the interviewee and interviewer present. A semi-structured interview guide (see File S1) was used for data collection, and it was pilot tested among members of the study group. The interview guide consisted of open-ended questions aimed at facilitating exploratory discussions [15,16]. Following an exploratory approach, the interviews began with broad and open questions to allow participants to feel comfortable and fully express their experiences [17]. Each participant took part in one interview, and all interviews were conducted in Danish. The interviews had a median duration of 30 min, ranging between 18 and 49 min. Data saturation was reached when no new information emerged in the interviews, as determined by the interviewers and in agreement with the analysts who both listened to and read the transcribed interviews [18]. The transcripts were not returned to the participants for correction or comments.

*2.3. Analysis and Findings*

The transcribed interviews were analyzed in Danish [19]. The analysis followed a structured approach using content analysis [20]. While reading the full-text transcriptions, notes and meaningful headings regarding the interview content were recorded in the margin to ensure a comprehensive description of all aspects. Labels were assigned to the notes and headings to categorize all possible categories. In addition, the content analysis aimed at identifying categories, two categories were predetermined based on the study´s objective: coping strategies and what late complications mattered the most to the patients. During the analysis process, an unexpected category emerged, which was "the period after discharge". This was placed as a subcategory (see Figure 1). The researchers involved in the analysis met to discuss the findings and reach consensus on the final categories. No specific software was used to do the analysis, and the participants were not invited to provide feedback on the findings.

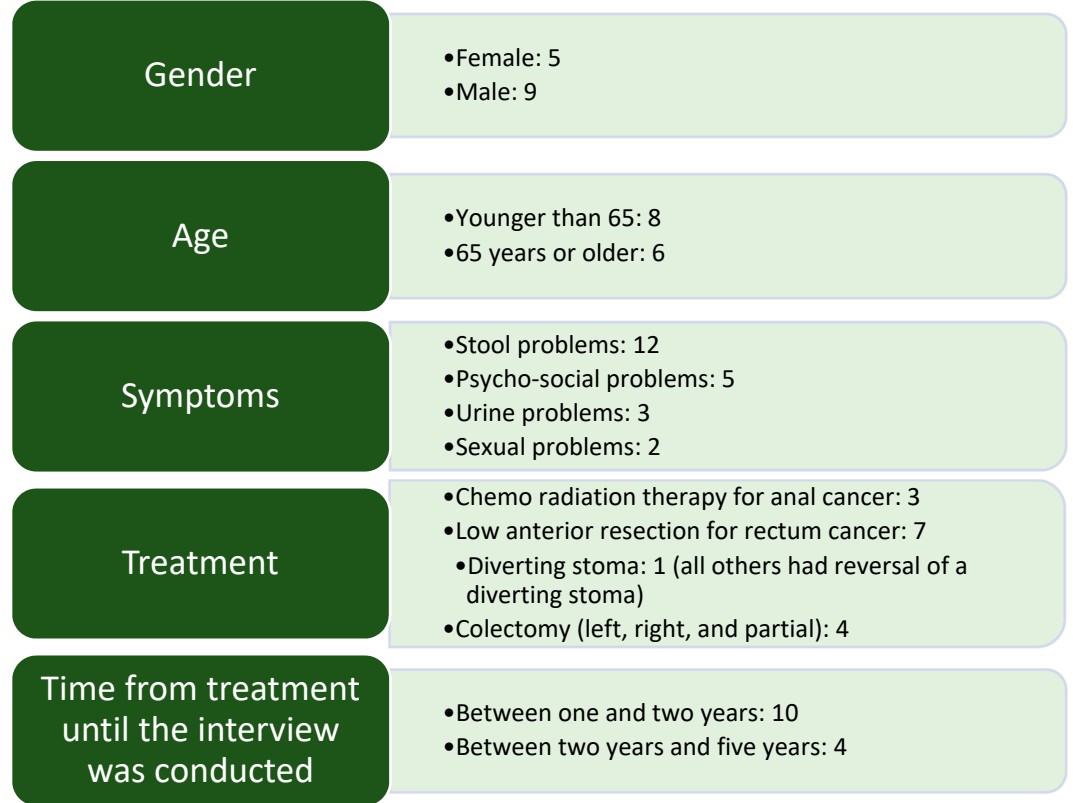

**Figure 1.** Demographic profile of the participants. Psycho-social problems refer to different non-specific issues adjusting to life circumstances after cancer treatment, such as being unable to work.

## 3. Results

We conducted a total of 14 interviews between June 2021 and August 2021. The demographic profile of the participants is shown in Figure 1. To secure the anonymity of the participants, we are not able to pin out the demographics of every single patient.

The participants presented with various symptoms, with many experiencing multiple symptoms, as seen in Figure 1. Our analysis identified two main categories and four sub-categories, as depicted in Figure 2.

**Figure 2.** Main categories and subcategories.

*3.1. How Patients React after Treatment for Colorectal and Anal Cancer*

Patients experienced an immediate reaction following discharge from surgery or completion of their last treatment of chemo-radiation therapy. Following this initial response, many patients developed various coping strategies to adapt to their post-cancer treatment life.

3.1.1. Period after Discharge

The patients were unaware of the magnitude of their journey until after they were discharged or had completed their chemo-radiation therapy. They could not have anticipated the challenges they would face prior to starting treatment. Surprisingly, many patients expressed that the cancer diagnosis itself was not the most difficult part; instead, the period following treatment was particularly challenging. Upon discharge, patients felt a profound sense of isolation. There was a stark contrast between the period before, when the cancer diagnosis was communicated, treatment plans were established, and medical care was provided, and the period after, where the patients were left to navigate their recovery largely on their own. Some patients felt they were discharged prematurely, with unresolved issues, e.g., difficulties with wound healing. The lack of guidance or information about the post-discharge period left patients feeling adrift, as though they were living in a vacuum of isolation.

> #10: "So there I was, done with all of that, and then you're sort of left to yourself again. You don't have Herlev (name of the Hospital) in the background anymore, so to speak."

3.1.2. Coping Strategies

In this study, all the patients experienced late complication symptoms. Prior to attending the Late Complication Clinic, the patients generally employed two coping strategies to manage these symptoms. Some chose to do nothing, while others turned to the internet for information.

Patients who sought advice for treatment on the internet often utilized platforms like Facebook, where they found specific groups dedicated to particular symptoms or diagnoses. They found these online communities to be immensely helpful, as they received advice from individuals who had experienced similar challenges. They acknowledge that much of the advice shared online was not evidence-based, but it still provided significant support. Others used Google to explore treatment options or engaged in self-diagnosis

to gain insights into their conditions. While they were aware that many of the treatment options lacked scientific evidence, they did not view it as a problem.

On the other hand, patients who did not use the internet expressed concerns about the overwhelming amount of information available and the confusion it caused. They found it challenging to determine what was accurate and reliable amidst the vast array of information. Additionally, some patients simply did not have the energy or capacity to search the internet for symptom information and treatment options.

> #1: *"I have Googled for advice myself, I know it's wrong to self-diagnose."*

Another coping strategy observed was to be open and discuss the challenges they were facing. They engaged in conversations with friends, family, and colleagues to share their experiences. They found that being open about their difficulties made it easier to maintain connections. They believed it was important not to conceal the difficult aspects because doing so would only increase their sadness. However, it should be noted that there were patients who had a different perspective. They preferred not to dwell on or discuss the negative aspects as they found it did not contribute to their well-being. Additionally, some individuals chose not to share their problems because they felt that people, in general, would not be interested in hearing about their struggles.

> #4: *"And I think it is important, you are honest with yourself—not least with yourself, but also with the people you are with, because then you get—it gives you peace of mind that—"well, they know very well that this is how . . . (name of the patient) is and he has to (go)—because of this and that and that". Then, I have to go. And I think it's important to be honest with the other people you're with and to say, well, that's just the way the situation is."*

### 3.2. Experienced Symptoms

In this study, patients primarily experienced stool problems as late complications. However, during the interviews, additional symptoms emerged, including fatigue and pain. These symptoms were reported by the participants and were not initially identified as part of their late complications but were revealed during the interview process.

### 3.2.1. Stool Symptoms

The patients in the study reported experiencing diarrhea and stool incontinence, both during the day and at night. They often had unpredictable bowel movements as they had lost the sensation and ability to control their stool. They frequently had bowel movements ranging from 3 to 15 times a day, including during the night. Many of them also described a sense of incomplete emptying. These stool-related issues consumed a significant amount of the patients' thoughts and attention. They felt tired and drained of energy; there was a constant awareness of stool, and some felt they had an odor of stool around them all the time. These challenges with bowel movements greatly impacted their daily lives, particularly when engaging in activities outside of their homes. It created difficulties in their social lives, leading many patients to avoid attending dinner parties, restaurants, cinemas, and theaters due to their bowel problems. Some patients even had catastrophic thoughts, such as the fear of being publicly embarrassed and having their face on the front page of the newspaper if they accidentally passed stool in a public place. Almost all the patients carried an extra bag containing diapers and spare clothing, and they had compiled a list of public toilets or locations where they could find accessible restroom facilities.

Patients in the study shared various strategies they employ to control their bowel movements. For example, one patient discovered that refraining from eating food from 6 p.m. until 12 a.m. helped reduce the likelihood of experiencing stool incontinence at work. Another patient found that cycling instead of walking reduced the risk of stool incontinence. Additionally, one patient observed that consuming starch instead of fiber resulted in a decrease in the number of bowel movements. These individualized approaches

highlight the patients' attempts to manage and mitigate their symptoms through dietary modifications and lifestyle adjustments.

Some patients in the study expressed that when they had a stoma, their lives were easier because they did not have to constantly think about finding a toilet. Additionally, they had received excellent support from stoma nurses.

On the other hand, patients who had undergone pelvic radiation therapy often faced challenges related to rectal bleeding. This had two significant implications. Firstly, there was a risk of blood staining their trousers, which could be embarrassing and distressing for the patients. Secondly, the rectal bleeding served as a reminder of their past experience with cancer, leading to a fear of recurrence.

> *#11: "I wear diaper during daytime, at night I wear women's pads, it is to absorb if I—what is it called—if I shit in my pants."*

### 3.2.2. Other Symptoms

Among the male patients in the study, complaints of sexual dysfunction were prevalent, while no female patients reported such issues. The primary complaint among male patients was erectile dysfunction. They explained that a nerve had been damaged during surgery, resulting in impotence. The patients expressed a desire for early information about the nerve damage from the surgeon following the surgery. Erectile dysfunction could manifest as a complete lack of erection or a partially affected weak erection.

Patients had experimented with various treatments, such as phosphodiesterase inhibitors or injections to address their erectile dysfunction. However, they were dissatisfied with the treatments for various reasons. Some patients desired a more natural solution, while others encountered barriers such as the need to refrigerate injections, which made them uncomfortable due to the possibility of others, including children, seeing them.

Patients also experienced difficulties during sexual intercourse. Some described a difference in sensation compared to before, and those without committed relationships found it challenging to initiate new relationships while dealing with erectile dysfunction. Patients with stomas reported a lack of sexual desire and refrained from engaging in sexual intercourse until after the diverting stoma was reversed.

These experiences underscore the significant impact of sexual dysfunction on the quality of life and intimate relationships of male patients in the study.

> *#12: "And I also have some dysfunction regarding erection, which is just arghhhh very annoying."*

In addition to the previously mentioned complications, the patients in the study also experienced urinary problems. Some reported a lack of sensation when urinating, while others experienced worsening urge incontinence following the surgery. The patients said that these issues were attributed to nerve damage during the surgical procedures.

A sense of weariness was commonly described by the patients. They expressed an inability to engage in activities as they did before cancer treatment. For instance, one patient could only work for four hours before feeling extremely tired for the remainder of the day.

Following radiation therapy, a patient reported experiencing pelvic and leg pain. The pain was severe and consumed her thoughts, making it difficult to focus on anything else. Pain management posed a challenge due to the impact of medications like morphine on bowel movements.

Another patient described a sensation of damaged abdominal muscles that would not heal, affecting his overall well-being.

These various symptoms and their debilitating effects highlight the diverse range of challenges faced by patients in coping with late complications after cancer treatment.

> *#5: "So I have developed some other late complications where I have been experiencing pelvic pain. They believe it's the radiation therapy that has caused it."*

## 4. Discussion

After being discharged from cancer treatment, patients often felt a sense of abandonment and being left to navigate their post-treatment journey on their own. In response, patients employed different strategies to seek help and support. Some turned to the internet, conducting Google searches or joining online peer groups on platforms like Facebook, to find information and guidance. However, the abundance of information available online could be confusing and unclear, leading some patients to feel overwhelmed and hesitant to rely on internet recommendations. Additionally, fatigue and exhaustion may hinder patients' motivation to actively seek help on the internet. Stool problems were a significant issue with a profound impact on daily life. Patients struggled with various complications. These problems constantly occupied their thoughts, making it challenging to engage in activities outside the home. Furthermore, it is important to note that late complications beyond stool problems also significantly affected patients' daily lives and overall quality of life. These complications can be wide-ranging, including urinary problems, fatigue, pain, sexual dysfunction, and musculoskeletal issues. The combined burden of these complications underscores the challenges faced by patients as they strive to maintain a sense of normalcy and well-being after cancer treatment.

The finding that patients often felt left on their own after discharge from cancer treatment is supported by other studies as well. Many patients describe experiencing a sense of limbo during this transitional phase. They no longer identify as cancer patients, but they also do not feel completely "normal" or back to their pre-cancer selves. This can lead to feelings of abandonment and isolation, as if everyone has moved on and left them behind. In fact, some patients even express anticipation and longing for visits to the hospital, as it provides them with a sense of connection and support [21]. These sentiments highlight the need for improved information and support regarding life after discharge from cancer treatment. Patients require clear guidance and resources to navigate this post-treatment phase successfully. Early follow-up visits or interventions at the hospital may also be necessary for certain patients to address their physical, emotional, and psychological needs during this critical period of adjustment. By providing patients with adequate information, support, and follow-up care after discharge, healthcare systems can help alleviate feelings of abandonment and improve the overall well-being and satisfaction of cancer survivors.

The observations in our study are in line with another study examining the perspectives of female patients on physicians' communication regarding sexual dysfunction. The study revealed that certain topics related to sexual dysfunction were not adequately addressed during consultations with physicians. As a result, patients felt the need to seek information and guidance on their own, often turning to the internet and online communities [22]. Patients reported that they had to navigate and determine potential treatment options for themselves, as they did not always have full confidence in their physicians' knowledge or willingness to discuss these sensitive topics. In contrast, online communities provided a judgment-free space where patients could freely discuss their experiences and gather valuable information about treatment options that were not readily provided within the hospital setting [22]. These findings suggest that there may indeed be alternative treatment options for late complications that are not necessarily evidence-based but are still beneficial to patients. However, these options may not be established or readily available within the traditional hospital setting. It highlights the importance of considering patient perspectives and exploring a more comprehensive approach to addressing late complications, including exploring alternative treatment modalities and incorporating patient experiences and preferences into clinical practice.

Our study is aligned with other studies that have examined patient willingness to discuss symptoms and feelings related to late complications. It is evident that there is variation among patients in terms of their openness and willingness to share their dilemmas and thoughts [21]. Some patients in our study, as well as in another study, expressed a reluctance to share their experiences, either stating that they were not the worrying type or that they did not experience any problems and could continue living

their lives as they did before the diagnosis and treatment of colorectal cancer [21]. This suggests that certain individuals may have a more resilient or self-reliant attitude, and they may not feel the need to openly discuss their concerns. On the other hand, another study found that patients were less willing to discuss late complications, particularly sexual problems, if the physician did not initiate the conversation or if they were not informed prior to treatment about the potential risks of such complications. Interestingly, in contrast to sexual problems, patients reported that physicians were more comfortable discussing stool problems [22]. This indicates that the willingness to discuss taboo or sensitive subjects may depend on various factors, including the physician's approach and the patient's perception of the legitimacy of discussing such topics. Overall, it is not clear whether some individuals genuinely do not experience problems due to their personality or coping style or if they would be open to discussing problems if given the opportunity and a safe space to do so. It highlights the importance of creating an environment that encourages open communication between patients and physicians, and where taboo subjects can be addressed without judgment or discomfort. Providing patients with adequate information through different media, like webpages, apps, or podcasts, about potential late complications and actively initiating discussions about these topics may help overcome barriers and facilitate meaningful conversations about patients' experiences and concerns.

Patients wanted information about nerve damage soon after surgery. As the nerves can be very difficult to identify during surgery [23], this information cannot be provided as the patients wish. Another study found that patients had the impression that the nerve damage was due to something having gone wrong during the surgery, and it seemed, to them, that it was purposefully not mentioned as a risk factor [22]. It is clear that surgeons need to inform their patients better or in other ways before and after pelvic surgery. Even though it is often not possible for the surgeon to tell if a nerve is damaged and to what extent, the patients should nevertheless be informed about the risk.

A strength of this study is the use of telephone interviews for data collection. Telephone interviews have previously been described as advantageous within qualitative research due to the ease of access/participation for the participants and researchers, thereby possibly easing recruitment to the study and reducing drop-outs [24]. Furthermore, patients might find that the physical distance between the interviewer and interviewee facilitates a sense of anonymity and comfort, making it easier to talk about sensitive or taboo subjects. Purposeful sampling of participants resulted in the recruitment of a diverse group of patients and, thereby, a sufficiently varied representation of the experiences of patients treated for colorectal and anal cancer. This, together with the rich data obtained during interviews, enhances the transferability and trustworthiness of the findings of this study. With the precise description of the method and the fact that we used more than one person to interpret the data have also enhanced the credibility and reflexivity of the study. The findings in the study are substantiated with findings from other studies and by patient citations, which increase the confirmability of the study. There are also limitations that should be considered. Follow-up interviews might have further elaborated or validated the categories that we defined as the main categories. The interviewers were all female, which might have affected how especially male participants expressed themselves about sensitive topics such as sexual dysfunction and feelings and concerns. We only conducted telephone interviews, and supplementary explorative methods such as patient diaries or focus group interviews could potentially have provided a deeper understanding of the everyday lives of these patients.

The clinical implications include the fact that these patients need more attention and better information than we are currently offering them. Late complications after treatment for colorectal and anal cancer are common and severe, and this study shows that patients experience being left on their own without any help from the established health care system. Insight into patients' coping strategies is important as it can inspire future interventions in surgical outpatient clinics and clinics with a special focus on late complications after

treatment for colorectal and anal cancer. This could be formal internet-based information, written information, face-to-face interventions, podcasts, etc. Furthermore, it would be interesting to test the treatment options found by patients on the internet for effectiveness and, if effective, implement them in the current treatment trajectory. Future research should investigate to what degree existing patient-reported outcome measures embrace the array of late complications uncovered in this study.

## 5. Conclusions

Patients felt left alone after discharge from the hospital after colorectal and anal cancer treatment. As a result, many patients sought and found help on online platforms. The patients suffered from severe stool problems, which hindered them from living a normal and active life and participating in social activities. Additionally, patients endured various other late complications, including urine leakage, pain, tiredness, and erectile dysfunction, all of which contributed further to their overall distress.

**Supplementary Materials:** The following are available online at https://www.mdpi.com/article/10.3390/curroncol30080546/s1, File S1: Interview guide about symptoms after treatment for colorectal and anal cancer.

**Author Contributions:** Conceptualization, B.T.O., J.R. and M.L.S.H.; formal analysis, B.T.O. and T.T.; investigation, M.L.S.H. and T.T.; writing—original draft preparation, B.T.O.; writing—review and editing, all authors. All authors have read and agreed to the published version of the manuscript.

**Funding:** This research received no external funding.

**Institutional Review Board Statement:** The study was approved by the Danish Data Protection Agency (P-2021-31), and the study was exempt from approval from the Committee on Health Research Ethics (Journal-no.: H-21020718). The study was conducted in full accordance with the Helsinki Declaration [25], and patients were included after obtaining informed consent to participate.

**Informed Consent Statement:** Informed consent was obtained from all the subjects involved in the study. The information about the interview was given either by telephone or by electronic communication to the patient's personal secure electronic mailbox (e-box). Furthermore, patients were informed about anonymity and confidentiality and about participation being voluntary and that withdrawal would have no consequences for their future treatment in the clinic. They were all informed that conversations were recorded.

**Data Availability Statement:** The data that support the findings of this study are available on request from the corresponding author. The data are not publicly available due to their containing information that could compromise the privacy of research participants.

**Acknowledgments:** We would like to acknowledge the valuable support and help from our dear colleague and nurse scientist, Anne Kjærgaard Danielsen, who died during the research process.

**Conflicts of Interest:** Birthe Thing Oggesen: Pharmanovia (honoraria to be in an advisory board) and PharmaCosmos (honoraria for lectures about late complications). Jacob Rosenberg: The author declares no conflict of interest. Marie Lousie Sjoedin Hamberg: The author declares no conflict of interest. Thordis Thomsen: The author declares no conflict of interest.

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
