# Peer review of "Exploring Patients’ Perspectives on Late Complications after Colorectal and Anal Cancer Treatment: A Qualitative Study"

_curroncol, doi:10.3390/curroncol30080546_

Round 1

Reviewer 1 Report

Exploring patients' perspectives on late complications after colorectal and anal cancer: a qualitative study

Oggesen et al. Current Oncology

The authors present a qualitative study based on telephone interviews with 14 patients who had received treatment for colorectal and anal cancer. The main conclusion is that patients felt left alone after hospital discharge and therefore a majority of them looked for and found help on various online platforms.

The topic is of major interest to any colorectal health care provider, and has historically been insufficiently investigated. The present study is therefore a sound initiative.

The parts on the period after discharge and regarding coping strategies are interesting and provide important knowledge. One important message of this manuscript is that a proportion of colorectal surgeons do not inform patients in a sufficient way regarding the panorama of potential complications, and especially sexual dysfunction.

The manuscript has a clear structure, is well written and the text is in general easy to follow.

Comments:

The authors mention a few facts regarding the study population such gender distribution and distribution < or > 65 years of age. With a limited study population the potential possibility of identifying individual patients from the manuscript must be considered, however this is not mentioned by the authors as a reason to give sparse information on the study population.

The present reviewer would suggest adding some information regarding the received treatment, for example the type of surgery performed (LAR with defunctioning stoma, LAR following reversal of defunctioning stoma, APR, Hartmann, anal cancer treated by radio-chemo and anal cancer treated by salvage surgery (if any such patients were included).

It would also be of interest to know how much time had passed from that treatment was completed to the time point of the interview.

The authors point out that all interviews were performed by two women and that this potentially may have affected male patients in a negative way when discussing sexual dysfunction. It would be interesting if the authors would discuss what they actually think regarding this potential weakness. If the authors were to perform a new similar study, would they consider to have the interviews done by a woman and a man together? Please explore.

The authors state that patients felt left alone following completed treatment. It would be of interest if the authors would describe the principles for follow up after surgery for colorectal cancer and after chemo-radiotherapy for anal cancer at their hospital, so that readers may relate to and evaluate their own practice.

The discussion regarding what information some of the patients found on internet is very interesting. Although speculative and perhaps difficult to answer, do the authors think the most important for the patients is to find useful information for their specific problems, or to find a group of people with whom they can relate and discuss their situation?

At hospital discharge, should the surgeon recommend the patient to look for internet platforms? And if so, any specific ones?

Minor comment: there seems to be some text missing following line 51.  

Reviewer 2 Report

Exploring patients' perspectives on late complications after colorectal and anal cancer: a qualitative study 

This explorative qualitativestudy aimed to investigate patients' experiences and coping strategies in relation to late complications after colorectal and anal cancer treatment. The comments are as follows:

1. The title needs to be further clarified: after colorectal and anal cancer treatment

2. Line 10, regarding”several measurement tools”, Please provide examples to illustrate

3. What specific time does late refer to?

4. Line50-51, the coping mechanism they employ cumulating evidence suggests that a significant minority.  what means?

5. Line45, regardingExisting scoring tools, Please give some examples

6. Introduction: Can you choose a theoretical framework to guide your research? For example, the theory of unpleasant symptoms, the theory of symptom management, or other theories that you are familiar with.

7. Line 79-92, Data were collected through telephone interviews, Didn't you have face-to-face interviews with the interviewees? But, with only the interviewee and interviewer present, why??? Please check and explain.

8. Please provide the inclusion and exclusion criteria for the patients.

9. Line 107-114, I suggest the authors provide a table.

10. Line 123, I suggest you refer to some literature on discharge readiness and redefine the sub themes”Period after discharge”

11. Line 140, I suggest the authors categorize and describe the coping strategies, For example, positive/negative coping strategies, adaptive/maladaptive... ...

12. Has the author included colorectal/anal cancer patients with permanent enterostomy? They also have many problems after discharge, and I suggest that you include them to obtain more new findings.

13. Please supplement the interview outline in the main text

14. Line 178, “ Experienced symptoms”, Please supplement a description of the psychosocial symptoms

15. Conflicts of Interest: Please keep the content concise

16. Line 93,  I suggest you use specific software to do the analysis in future qualitative study, e.g., NVIVO, Dedoose... ...

Minor editing 

Reviewer 3 Report

I would like to congratulate the authors on their fascinating work regarding this interesting qualitative study on exploring patients' perspectives on late complications after colorectal and anal cancer. The manuscript is well-written

I strongly recommend acceptance for publication of the paper after revision.

1) I would suggest a brief discussion on the role of sepsis after colorectal operations

This complication has been associated with negative economic impact, increased morbidity, extended postoperative hospital stay, and death. I would suggest adding this important information to the discussion section and consider citing the recently published articles

https://pubmed.ncbi.nlm.nih.gov/35371356/
https://www.ncbi.nlm.nih.gov/pmc/articles/PMC1950493/
https://www.ncbi.nlm.nih.gov/pmc/articles/PMC6085193/

2) How did the patients in your study experience this complication?

3) I would suggest adding a picture of your questionnaire to the patients

4) Are there recent quantitative articles related to this topic? Do they agree with the results of your study?

Reviewer 4 Report

The authors present a qualitative interview study in which they investigate how, the patients undergoing colorectal cancer treatments, manage late complications.

The topic is interesting, relevant and the design of the study is original.

The article is well written, but the research methodology is somewhat weak.

The results are not properly presented. The discussion is well written but need to be supplemented and integrated. The limitations of the study are discussed. The conclusions are appropriate.

There are some problems to be solved:

1)    The research structure of the study is lacking; the authors should present the results they obtained as table with number and percentage.

2)    The study is lacking some important data as surgical data (which operation the patients have undergone? Protective ileostomy or not? This data are important for the aim of the study

3)    The discussion should be focused also about the main problems of colo-rectal cancer treatment:

-       Impact of chemoradiotherapy on LARS, urinary symptoms, sexuals problems ecc

-       Impact of surgery on post-operative late complications

-       Method to prevent these complications (future perspectives) as: icg guided surgery, minimally invasive surgery, management of protective ileostomy

For the discussions, I recommend some important studies who point out the attention on these problems: 

Runkel N, Reiser H. Nerve-oriented mesorectal excision (NOME): autonomic nerves as landmarks for laparoscopic rectal resection. Int J Colorectal Dis. 2013 Oct;28(10):1367-75. doi: 10.1007/s00384-013-1705-x. Epub 2013 May 11. PMID: 23666512.

Brescia A, Muttillo EM, Angelicone I, Madaffari I, Maggi F, Sperduti I, Gasparrini M, Osti MF. The Role of Indocyanine Green in Laparoscopic Low Anterior Resections for Rectal Cancer Previously Treated With Chemo-radiotherapy: A Single-center Retrospective Analysis. Anticancer Res. 2022 Jan;42(1):211-216. doi: 10.21873/anticanres.15475. PMID: 34969727.

Round 2

Reviewer 3 Report

The majority of the requested changes were addressed accordingly. It can be accepted for publication without further changes in my opinion.